# Phylogenetic and Functional Structure of Wood Communities among Different Disturbance Regimes in a Temperate Mountain Forest

**Peikun Li** [1,2]**, Zihan Geng** [1,2]**, Xueying Wang** [3]**, Panpan Zhang** [1,2]**, Jian Zhang** [1,2]**, Shengyan Ding** [1,2,*] **and Qiang Fu** [4,*]

1. Key Laboratory of Geospatial Technology for the Middle and Lower Yellow River Regions, Ministry of Education, Kaifeng 475004, China; peikunlee@126.com (P.L.); 104753190167@henu.edu.cn (Z.G.); zhangpanpan@henu.edu.cn (P.Z.); zj_0305@henu.edu.cn (J.Z.)
2. College of Environment and Planning, Henan University, Kaifeng 475004, China
3. Henan Forestry Investigation and Planning Institute, Zhengzhou 450002, China; wangsnow0909@163.com
4. College of Landscape Architecture and Art, Henan Agricultural University, Zhengzhou 450002, China
* Correspondence: syding@henu.edu.cn (S.D.); fqlandscape@126.com (Q.F.); Tel.: +86-0371-23881102 (S.D.)

**Abstract:** The mechanisms responsible for biodiversity formation and maintenance are central themes in biodiversity conservation. However, the relationships between community assembly, phylogeny, and functional traits remain poorly understood, especially following disturbance. In this study, we examined forest community assembly mechanisms in different disturbance regimes across spatial scales and including tree life history classes, using phylogenetic and functional trait metrics. Across disturbance regimes, phylogenetic structure tended to be over-dispersed, while functional structure tended to be clustered. The over-dispersion of phylogenetic structure also increased from small to large diameter species. Moreover, the explanation of spatial distance for the turnover of phylogenetic and functional structure was increased, while environmental distance explained less structure as disturbance intensity decreased. Our findings suggest that niche theory largely explains forest community assembly in different disturbance regimes. Furthermore, environmental filtering plays a major role in moderate to high disturbance regimes, while competitive exclusion is more important in undisturbed and slightly disturbed ecosystems.

**Keywords:** net relatedness index; functional trait; niche theory; habitat filtering; competitive exclusion; deciduous broad-leaved forests





## 1. Introduction

Biodiversity formation and maintenance mechanisms, and community assembly mechanisms in particular, are central themes in biodiversity conservation [1]. Niche theory holds that the niche differentiation among coexisting species strongly affects community construction, which results from forces including habitat filtering and competitive exclusion [2,3]. In contrast, neutral theory posits that stochastic factors, such as diffusion and random effects, are the determinants of community construction [4]. Based on the phylogenetic niche conservation theory of Webb [5], the phylogenetic distance of species within communities can be used to infer the relative strengths of niche and neutral progress in community assembly. If the evolution of species functional traits is relatively slow, habitat filtering is predicted to lead to clustered community phylogenetic structures, while competitive exclusion leads to over-dispersed communities [6–9]. Random phylogenetic structures may result from diffusion and habitat filtering or a combination of random effects and competitive exclusion [10,11]. Community functional trait structure, therefore, represents a comprehensive pattern of species functional traits [12]. The existing community trait distributions result from differences in the selection of environmental and non-environmental factors by species, and thus provide important clues to understanding the relative importance of ecological

processes in community construction [13,14]. With improved phylogenetic and functional ecology methods, community phylogenetic studies of plant functional traits have become common tools for assessing community construction mechanisms. The roles of niche and neutral processes in community construction, based on phylogenetic or functional traits and $\alpha-$ and $\beta-$diversity, have attracted much attention [15–19]. Studies have shown that the $\alpha$- and β-diversity of community phylogenetic and functional traits are closely related to study scales, both in time and space [20,21]. Interspecific interactions and diffusion restrictions are more prevalent at smaller community scales, while environmental filtering is generally a feature of larger scales [22,23]. Meanwhile, the α-diversity of community phylogenetic and functional traits also show different response patterns for tree species at different life history stages, due to different environmental needs and tolerances [20]. For example, small and medium diameter tree species are commonly subject to habitat filtering, leading to clustered community phylogenetic structures [24]. Whereas, competitive exclusion is more likely to occur between large diameter trees, due to the need for more resources, resulting in over-dispersion [9]. The relative importance of diffusion and environmental filtering in community construction can be inferred from changes in the phylogenetic signals and ecological characteristics of species between communities [25,26]. Although phylogenetic and functional trait diversity is increasingly used to infer community assembly mechanisms individually, most studies do not combine them [27].

Disturbances, from human activities to natural fires and earthquakes, have profound impacts on regional community construction and species diversity [28]. With the increasing frequency of human activities, human disturbance has become the primary factor affecting the construction of regional communities [29]. The influence of human disturbance on ecosystems has long been a focus of multidisciplinary research in geography, ecology, and natural resources science [30]. Disturbance theory is a vital part of ecology, and the "intermediate-disturbance hypothesis" is currently the most studied [31,32]. This hypothesis suggests that moderate disturbances help maintain high biodiversity [33,34]. Generally, unmanaged forests after human disturbance are in the early and middle stages of succession [35]. However, some extreme disturbances can reverse the succession of secondary forests, which seriously threatens healthy forest development [36].

Deforestation is among the most common human disturbances [37] and affects forest phylogenetic and functional trait structure, and subsequently alters forest community biodiversity and ecosystem function [11]. Differences in the biotic (e.g., community structure and species composition) and abiotic (e.g., soil and light) environments [38–42] resulting from different deforestation methods, intensities, and intervals lead to different effects on the structure, function, and biodiversity of forest ecosystems [43]. Therefore, studying the effects of deforestation disturbance on regional community construction and species diversity is of great significance for the renewal and development of forest communities [44]. Most studies have focused on the impact of disturbances on forest community structure, stability, and species diversity [29,44–48]. However, there are few studies of community assembly mechanisms that examined the phylogenetic and functional trait structure of woody plants as succession progressed.

In this study, we examined the community assembly mechanisms of woody plants in forests subject to different disturbance regimes, at spatial and diameter at breast height (DBH) scales, using phylogenetic signals and the phylogenetic and functional trait structure. We hypothesized that (1) species with similar genetic relationships would have similar functional traits, as a result of significant phylogenetic signals; (2) the phylogenetic and functional trait structure at small and medium spatial and DBH class scale could have higher clustering, due to habitat filtering and competitive exclusion; and (3) environmental filtering could tend to be more important following high and moderate disturbance, due to increased resource availability and species richness, and competitive exclusion could tend to be more important for slightly disturbed and undisturbed communities, due to diffusional limitation.

## 2. Materials and Methods

### 2.1. Study Site and Sampling

The study site was located in the Baiyun Mountain National Nature Reserve (111°48′–112°16′ E, 33°33′–33°56′ N), Luoyang, south of Henan Province, China (Figure 1), which is about 168 km² and 1500–2216 m above sea level [29]. The slope of the mountain is mostly 40–80°. Long-term mean annual precipitation is approximately 1200 mm, with most occurring from July to September, and the mean annual relative humidity is 70–78%. Mean annual temperature was 13.1–13.9 °C, and extreme minimum and maximum temperatures were −14.4 and 42.1 °C, respectively. The soil texture is mainly light soil, with a pH of 5.5–6.5 [49].

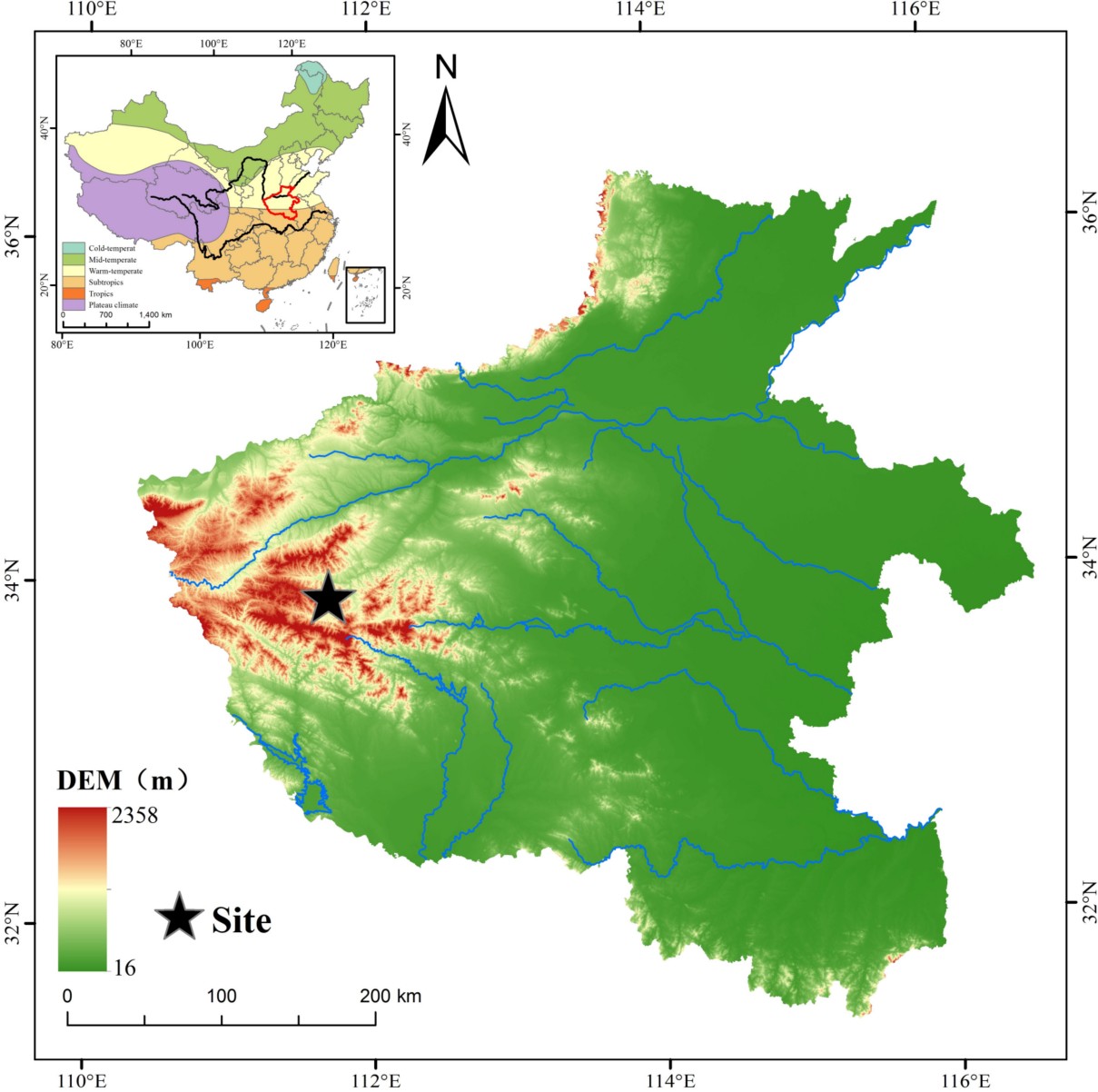

**Figure 1.** Location of different disturbance dynamics plots in the temperate−subtropical ecological transition zone of the Baiyun Mountain Nature Reserve. DEM represents altitude.

The Baiyun Mountain Nature Reserve is located in a temperate–subtropical ecological transition zone, with mostly deciduous broad-leaved forests. The forest coverage in the reserve reaches 98.5%, which consists of 1991 species of plants, including the following

dominant species: *Quercus aliena* var. *acutiserrata*, *Carpinus turczaninowii*, *Betula platyphylla*, *Pinus armandii Franch*, and *Toxicodendron vernicifluum* [49].

Forest monitoring plots were randomly selected and stratified by disturbance regime in the Baiyun Mountain Nature Reserve, where plant growth and ecosystem functions are sensitive to climate change [50]. Four disturbance regimes of the forest were estimated, based on knowledge of local logging events and forest physiognomy. Four 1 hm$^2$ plots (100 m × 100 m), namely, plantation, twice-cut, once-cut, and old-growth forests, were randomly selected within each disturbance regime in the reserve (Table 1). Four 1 hm$^2$ plots were divided into 100 grids (10 m × 10 m). All trees with diameter at breast height (DBH) ≥1 cm in the plot were tagged, mapped, and measured [51]. Topographic variables (elevation, convexity, slope, and aspect) were measured using the methodology of Harms [52] for each 10 m × 10 m grid in the plot.

**Table 1.** Summaries of disturbance regime forest plots in Baiyunshan Nature Reserve.

| | Plantation Forest | Twice-Cut Forest | Once-Cut Forest | Old-Growth Forest |
|---|---|---|---|---|
| Average elevation (m) | 1647.4 | 1578.66 | 1413.15 | 1772.62 |
| Mean DBH (cm) | 14.23 | 7.21 | 7.76 | 9.5 |
| Total basal area (m$^2$) | 22.925 | 25.95 | 33.33 | 31.9 |
| Number of species | 42 | 46 | 57 | 52 |
| Individual number | 953 | 2534 | 3671 | 2318 |
| Disturbance regimes | *Larix kaempferi* forest planted after logging and clearing. | Natural regeneration occurred after once-cutting. Twice-cutting and breeding were carried out after about 30 years natural recovery, followed again by natural recovery. | The forest was restored after comprehensive once-cutting. | The forest has been undisturbed for more than 100 years. |
| Age of forest (years) | 20 | 50 | 50 | 100 |
| Degree of disturbance | High disturbance | Moderate disturbance | Slight disturbance | Undisturbed |
| Dominant species | *Quercus aliena* var. *acutiserrata*; *Larix gmelinii* | *Quercus aliena* var. *acutiserrata*; *Pinus armandii Franch*; *Corylus heterophylla* | *Quercus aliena* var. *acutiserrata*; *Pinus armandii Franch*; *Forsythia suspensa* | *Quercus aliena* var. *acutiserrata*; *Sorbus hupehensis*; *Litsea tsinlingensis* |

To assess the relationship between phylogenetic structure and spatial scale, we further divided each 1 hm$^2$ plot into 10 m × 10 m, 20 m × 20 m, and 25 m × 25 m grids, with a total of 100, 25, and 16 of each size, respectively. To investigate the relationship between time scale (as measured by tree size) and phylogenetic structure, we divided all woody plants with a DBH ≥1 cm into three different diameter classes following [53]; namely, small (1 cm ≤ DBH < 5 cm), medium (5 cm ≤ DBH < 10 cm), and large (DBH ≥ 10 cm).

## 2.2. Phylogenetic Tree Construction

Phylogenetic trees were constructed using the database Phylomatic [8]. All species information in each plot was imported into the community phylogenetic software Phylocom Version 3.21 (available online http://www.phylodiversity.net/phylocom, accessed on 4 February 2018) [54]. First, major relationships were taken from the Angiosperm Phylogeny Group classification (APG IV 2016). Second, the BLADJ algorithm was implemented within Phylocom to calibrate each species pool supertree, by applying known molecular and fossil dates [55] to nodes on the supertree, resulting in ultrametric phylogenetic trees of each community (Figure 2).

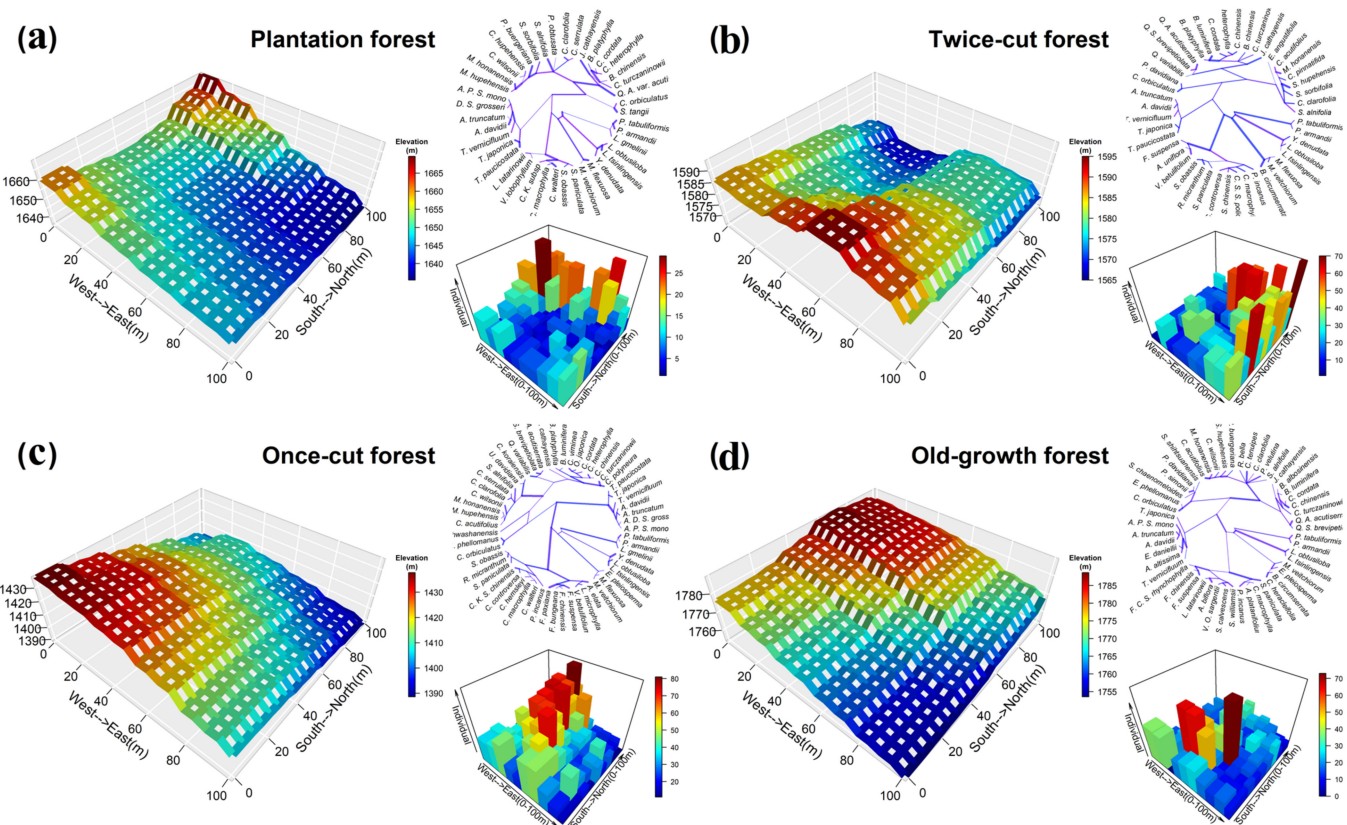

**Figure 2.** Topographic maps, phylogenetic trees, and spatial species abundances of woody plants in the plantation (**a**), twice-cut (**b**), once-cut (**c**), and old-growth (**d**) forests of the Baiyun Mountain Nature Reserve.

### 2.3. Functional Trait Clustering

We measured seven key functional traits representing the leaf (specific leaf area, stomatal conductance), stem (maximum tree height, wood density), and physiological (minimal fluorescence, non−photochemical quenching, transpiration rate) traits of the tree species in the community, according to the handbook for standardized measurement of plant functional traits by Pérez−Harguindeguy [56]. We reduced the dimensionality of these traits through principal component analysis (PCA). The first three axes were selected as comprehensive functional trait factors to transform the trait matrix into a distance matrix, and hierarchical clustering was conducted according to the trait distances between species, to generate a functional trait clustering tree [57].

### 2.4. Data Analysis

#### 2.4.1. Phylogenetic Signal

The phylogenetic signal was analyzed using Blomberg's $K$ [58], which is a measure of the observed trait variance compared to that expected under Brownian motion. The null expectation of $K = 0$ represents no phylogenetic signal, while $K = 1$ indicates a strong phylogenetic signal, and that the trait evolves according to Brownian motion. A weak phylogenetic signal is indicated by $0 < K < 1$, whereas $K > 1$ indicates a very strong signal, and that the trait values are more similar than expected under Brownian motion. The significance of the phylogenetic signal can be obtained by comparing variance observations of standardized independent differences across the phylogenetic tree for functional traits with a random test of the null model.

### 2.4.2. Community Phylogenetic and Functional Trait Structure

The net relatedness index (*NRI*) and standardization mean pairwise trait distance (*S.E.S PW*) were calculated to reflect the phylogenetic and functional trait character structures, respectively, of tree species in each spatial scale and diameter class, for each disturbance regime [5,59]. First, the mean phylogenetic distances (*MPDs*) and mean pairwise trait distances (*PWs*) for all species pairs in the quadrat were quantified. Then, we used a richness null modelling approach to estimate the expected subplot species richness distributions under random processes; we randomly permuted the species set of the phylogenetic tree or functional trait clustering tree 999 times to obtain the *MPD* or *PW* of each species pair in the quadrat under the random null model [60]. Finally, the observed values were normalized using the random distribution result, to obtain the values of *NRI* and *S.E.S PW*, calculated using the following formula [5]:

$$NRI = -1 \times \frac{(MPD_{obs} - \mathrm{mean}(MPD_{rnd}))}{\mathrm{sd}(MPD_{rnd})}, \tag{1}$$

$$S.E.S \; PW = -1 \times \frac{(PW_{obs} - \mathrm{mean}(PW_{rnd}))}{\mathrm{sd}(PW_{rnd})} \tag{2}$$

where $MPD_{obs}$ and $PW_{obs}$ represent the observed *MPD* and *PW* values; $MPD_{rnd}$ and $PW_{rnd}$ represent the *MPD* and *PW* values of 999 randomly generated null communities; and sd $(MPD_{rnd})$ and sd $(PW_{rnd})$ are the standard deviations of the 999 $MPD_{rnd}$ and $PW_{rnd}$ values, respectively. Negative values of *NRI* and *S.E.S PW* indicate higher mean phylogenetic distances and mean pairwise trait distance, respectively, than expected given the random assemblages, and are indicative of phylogenetic and functional trait over-dispersion. Whereas positive *NRI* and *S.E.S PW* values indicate lower mean distances and phylogenetic and functional trait clustering, respectively.

Previous phylogenetic studies have shown that the distributions of *NRI* and *S.E.S PW* scores from multiple equally sized quadrats are generally right-biased [9]. Therefore, we used the nonparametric Wilcoxon test to test for significant deviations between *NRI* or *S.E.S PW* and zero [61]. Moran's I was used to test the spatial autocorrelation of species *NRI* and *S.E.S PW* at different scales [62]. Spatial autoregression analyses (SAR) were used to analyze the effects of removing spatial autocorrelation on community phylogenetic structure and functional trait structure [54].

### 2.4.3. Beta Diversity of Phylogenetic and Functional Traits

The mean pairwise distance ($D_{pw}$) index was used to measure the phylogenetic or functional dissimilarity among the four disturbance regimes at different scales [63]:

$$D_{pw} = \frac{\sum_{i=1}^{n_{k1}} f_i \overline{\delta_{ik2}} + \sum_{j=1}^{n_{k2}} f_j \overline{\delta_{jk1}}}{n_{k1} + n_{k2}} \tag{3}$$

where $\overline{\delta_{ik2}}$ is the mean pairwise phylogenetic or pairwise trait distance between species *i* in community $k_1$ to all species in community $k_2$ and $\overline{\delta_{ik1}}$ is the mean pairwise phylogenetic or pairwise trait distance between species *j* in community $k_2$ to all species in community $k_1$; and $f_i$ and $f_j$ are the relative abundances of species *i* and species *j*, respectively.

The four disturbance regime forests were divided into 20 m × 20 m subplots, and the Euclidean distances between the centers of the 25 subplots in each plot were calculated as a spatial distance. Environmental distances were measured as the Euclidean distances between environmental factors (standardize slope, aspect, elevation, and convexity) to create a standardized environmental matrix. We calculated $D_{pw}$ values between the 100 quadrats and used Mantel tests to measure the correlations between $D_{pw}$ and environment matrices. Multiple regression on distance matrices (MRM) was used for the partial Mantel tests of spatial distance, environmental distance, and $D_{pw}$. The MRM was used to decompose the variance of the phylogenetic β-diversity value into three parts: spatial distance, environ-

mental distance, and the interaction between the two. MRM was used to assess the effects of spatial and environmental distance on community phylogenetic and functional trait turnover [64].

Phylogenetic and functional indices, Blomberg's *K*, and associated *p*-values were estimated with the "picante" package [25]. Moran's I and spatial autoregression analyses were conducted with the "spdep" package [64]. Mantel tests and MRM were conducted with the "ecodist" package [64]. All statistical analyses were conducted in R 3.4.0 (R Development Core Team, http://www.Rproject.org, accessed on 4 February 2018) [65].

## 3. Results

### 3.1. Phylogenetic Signals

Across the four disturbance plots, we detected phylogenetic signals ($K > 0$, $p < 0.05$) for all traits, except transpiration rate (TR), in the twice-cut forest (Table 2). Blomberg's *K* was smallest for maximum tree height (MTH) and greatest for non-photochemical quenching (NPQ), ($K > 1$ in twice-cut and old-growth plots). Therefore, the evolutionary history explained much of the functional trait variation of the plant species in the Baiyun Mountain plots; that is, species with similar kinship had similar functional traits.

**Table 2.** Phylogenetic signal as measured by Blomberg's *K* of functional traits in four disturbance regimes. MTH = maximum tree height (m); SLA = specific leaf area ($cm^2 \cdot g^{-1}$); WD = wood density ($g \cdot cm^{-3}$); F0 = minimal fluorescence; NPQ = non-photochemical quenching; TR = transpiration rate ($mol \cdot m^{-2} \cdot s^{-1}$); SC = stomatal conductance ($mmol \cdot m^{-2} \cdot s^{-1}$). "*" and "**" represent $p < 0.05$ and $p < 0.01$, respectively.

| Trait | Plantation | | Twice-Cut | | Once-Cut | | Old-Growth | |
|---|---|---|---|---|---|---|---|---|
| | *K* | *p* | *K* | *p* | *K* | *p* | *K* | *p* |
| MTH | 0.376 | 0.007 ** | 0.27 | 0.035 * | 0.248 | 0.02 * | 0.406 | 0.001 ** |
| SLA | 0.649 | 0.001 ** | 0.531 | 0.001 ** | 0.686 | 0.001 ** | 0.616 | 0.001 ** |
| WD | 0.462 | 0.037 * | 0.593 | 0.039 * | 0.425 | 0.049 * | 0.617 | 0.003 ** |
| F0 | 0.899 | 0.001 ** | 0.662 | 0.022 * | 0.918 | 0.001 ** | 0.702 | 0.001 ** |
| NPQ | 0.968 | 0.004 ** | 1.206 | 0.005 ** | 0.915 | 0.003 ** | 1.098 | 0.004 ** |
| TR | 0.499 | 0.017 ** | 0.515 | 0.074 | 0.474 | 0.02 * | 0.588 | 0.005 ** |
| SC | 0.75 | 0.001 ** | 0.387 | 0.004 * | 0.394 | 0.001 ** | 0.318 | 0.029 * |

### 3.2. Phylogenetic and Functional Structure at Spatial and DBH Scales

The phylogenetic structure tended to be over-dispersed across disturbance regimes, both overall and at different spatial scales and diameter classes (Table 3). Specifically, we observed over-dispersion (*NRI* < 0, $p < 0.05$) in all four disturbance regimes overall and with large DBH species, twice-cut and once-cut plots with medium DBH species, and in the twice-cut plot with small DBH species (Figure 3a, Table 3). Moreover, we observed over-dispersion in the once-cut plots at 20 × 20 m and in medium diameter DBH species at 10 × 10 m and 20 × 20 m.

Within DBH classes we found evidence of a clustered functional structure in different disturbance regimes. With the exception of the plantation plot with all DBH species and the once-cut plot with medium DBH species, we detected functional clustering (*S.E.S PW* > 0, $p < 0.05$) at 10 × 10 m, 20 × 20 m, and 25 × 25 m scales across disturbance regimes with overall, small, and medium DBH (Figure 3b, Table 3). However, the *NRI* in plantation and once-cut plots with large DBH species at 10 × 10 m and 20 × 20 m scales were functionally over-dispersed.

**Table 3.** Results of *t*−test for the hypothesis that the mean values of *NRI* and *S.E.S PW* is zero at different spatial scales and DBH classes in four disturbance regimes. D1, D2, D3, and D4 represent the plantation, twice-cut, once-cut, and old-growth forests, respectively. "*", "**", and "***" represent $p < 0.05$, $p < 0.01$, and $p < 0.001$, respectively.

| Space | DBH | NRI | | | | S.E.S PW | | | |
|---|---|---|---|---|---|---|---|---|---|
| | | D1 | D2 | D3 | D4 | D1 | D2 | D3 | D4 |
| 10 × 10 m | Overall | 3.969 *** | 7.428 *** | 26.777 *** | 6.159 *** | 1.354 | 17.931 *** | 6.93 *** | 5.965 *** |
| | Small | 0.415 | 1.589 | −2.516 * | −1.186 | 2.429 * | 14.449 *** | 10.541 *** | 3.507 *** |
| | Medium | 1.102 | −2.43 * | 18.422 *** | −3.018 ** | 3.88 *** | 9.53 *** | −0.117 | 4.419 *** |
| | Large | −2.635 * | 6.847 *** | 18.515 *** | −3.302 ** | −2.542 * | 11.116 *** | −3.188 ** | 1.453 |
| 20 × 20 m | Overall | 6.677 *** | −5.24 *** | 16.486 *** | 5.764 *** | 0.553 | 9.396 *** | 3.1 ** | 5.332 *** |
| | Small | 0.432 | −0.343 | −3.503 ** | −2.052* | 2.897 * | 11.695 *** | 9.0 *** | 3.456 ** |
| | Medium | −0.239 | −2.64 * | 15.024 *** | −2.256* | 2.572 * | 7.745 *** | −0.732 | 3.376 ** |
| | Large | 5.179 *** | 3.863 *** | 12.809 *** | −3.033 ** | −2.19 * | 5.319 *** | −3.154 ** | 1.437 |
| 25 × 25 m | Overall | 6.499 *** | 4.947 *** | 10.141 *** | 4.309 *** | 1.535 | 9.661 *** | 3.912 ** | 3.995 ** |
| | Small | 0.192 | −0.352 | −3.74 ** | −1.449 | 4.341 *** | 14.522 *** | 7.311 *** | 2.656 * |
| | Medium | −0.124 | −2.631 * | 14.267 *** | −1.973 | 3.155 ** | 6.039 *** | 0.045 | 2.711 * |
| | Large | 4.336 *** | 6.393 *** | −8.01 *** | −2.232* | −1.845 | 4.5 *** | −1.943 | 1.294 |

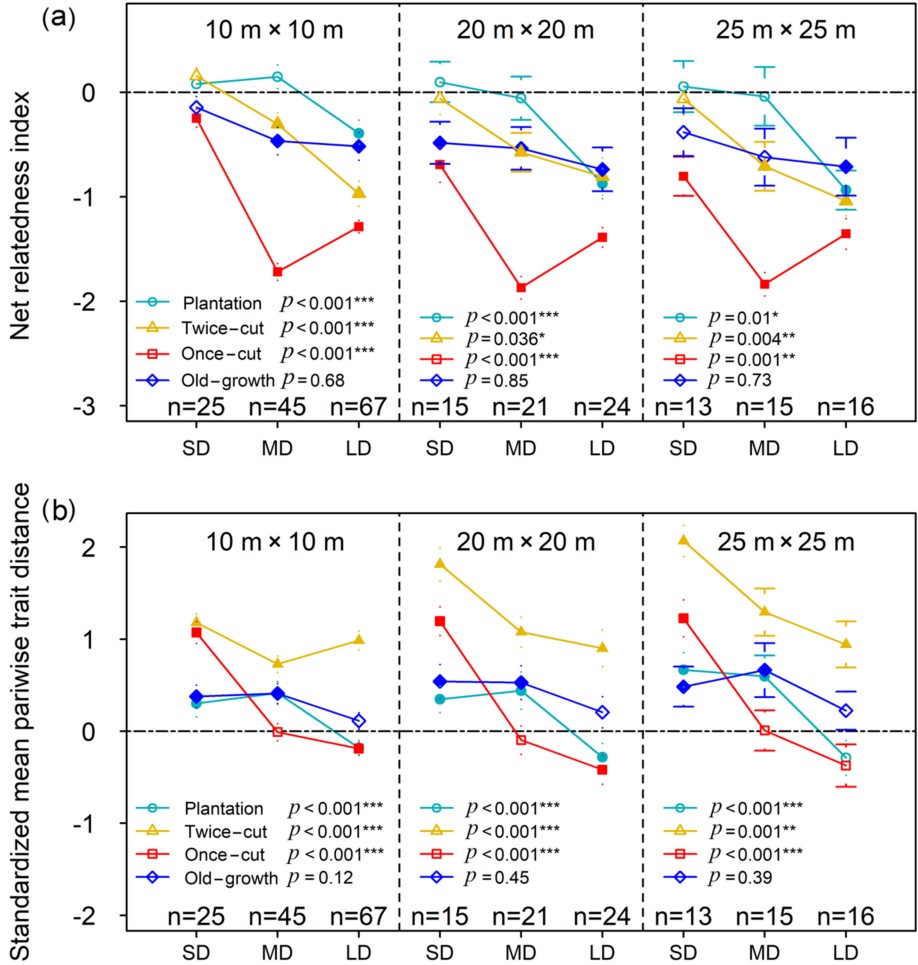

**Figure 3.** Phylogenetic (**a**) and functional trait structure (**b**) (mean ± SE) of different diameter classes in four disturbance plots at three spatial scales. Solid markers represent means of *NRI* or *PW* that are significantly different from 0 and the open markers represent non-significant differences based on the Wilcoxon test. "*", "**", and "***" represent $p < 0.05$, $p < 0.01$, and $p < 0.001$, respectively.

The communities tended to be more phylogenetically over-dispersed as DBH class increased ($p < 0.05$, Figure 3a), while also shifting from functional clustering to functional randomness, and even over-dispersion ($p < 0.05$, Figure 3b). With respect to spatial scale, we found that the phylogenetic and functional structure was relatively scale-independent within DBH classes, as we detected few significant differences between scales ($p > 0.05$, Figure S1). However, the phylogenetic structure decreased significantly in once-cut forests with overall and small DBH species ($p < 0.05$, Figure S1a,b), and the functional structure increased significantly in twice-cut forest in overall, small, and medium DBH species ($p < 0.05$, Figure S1c,d).

### 3.3. Phylogenetic and Functional Structure in Different Disturbance Regimes

All disturbance regimes exhibited over-dispersion of the overall DBH class, and the once-cut community was significantly more over-dispersed than the plantation and twice-cut communities ($p < 0.05$, Figure 4a). Within DBH classes, the once-cut community showed higher over-dispersion than more disturbed communities in the small DBH class ($p < 0.05$, Figure 4c), and the highest over-dispersion in the medium and large DBH classes ($p < 0.05$, Figure 4e,g).

In the overall DBH class, the functional structure trended to be clustered in twice-cut, once-cut, and old-growth communities, but was random in the plantation forest (Figure 4b). The twice-cut community had the strongest clustering in all DBH classes ($p < 0.05$, Figure 4b,d,f,h). Functional clustering tended to decrease with increasing disturbance in small diameter species, although the plantation (most disturbed) showed a clustering similar to the old-growth forest ($p < 0.05$, Figure 4d).

### 3.4. Beta Diversity of Community Phylogenetic and Functional Traits

The turnover in phylogenetic and functional traits was generally non-random in each disturbance regime and across DBH classes, as measured by *S.E.S. $D_{pw}$* ($p < 0.05$, Table 4). At the overall DBH level, the plantation ($2.33 \pm 0.62$) and once-cut ($2.364 \pm 0.51$) communities had the largest phylogenetic *S.E.S. $D_{pw}$* and the plantation community had the largest functional *S.E.S. $D_{pw}$* ($1.458 \pm 0.41$). The phylogenetic *S.E.S. $D_{pw}$* of the small DBH species was consistently the smallest across DBH classes and in different disturbance regimes (Table 4). Compared with the null-model, the observed phylogenetic and functional traits varied more rapidly than expected across subplots at all scales. Both the phylogenetic and functional turnover between paired plots was greater than zero ($p < 0.05$, Table 5), and the small DBH species had the lowest turnover (Table 5, Figure 5).

**Table 4.** Phylogenetic and functional standardized mean pairwise distances (mean *S.E.S. Dpw* $\pm$ SE) between disturbance communities. SD, MD, and LD represent small, medium, and large diameter classes, respectively. "*", "**" and "***" represent $p < 0.05$, $p < 0.01$, and $p < 0.001$, respectively.

| | | Plantation | Twice-Cut | Once-Cut | Old-Growth |
|---|---|---|---|---|---|
| *S.E.S. $D_{pw}$* of phylogenetic | Overall | 2.33 ± 0.62 *** | 1.156 ± 0.63 *** | 2.364 ± 0.51 *** | 1.541 ± 0.93 *** |
| | SD | 0.197 ± 0.97 *** | 0.513 ± 0.81 ** | 1.521 ± 0.82 *** | 1.007 ± 1.08 *** |
| | MD | 0.576 ± 1.3 ** | 0.915 ± 0.91 ** | 2.481 ± 0.37 *** | 1.037 ± 1.15 *** |
| | LD | 1.599 ± 0.65 *** | 1.285 ± 0.57 *** | 2.529 ± 0.57 *** | 1.073 ± 1.25 *** |
| *S.E.S. $D_{pw}$* of functional traits | Overall | 1.458 ± 0.41 *** | 0.414 ± 0.6 * | 0.611 ± 0.55 ** | 0.878 ± 0.59 ** |
| | SD | 0.209 ± 1.03 ** | −0.265 ± 0.54* | −0.407 ± 0.92 ** | 0.208 ± 0.56 * |
| | MD | 0.737 ± 0.9 ** | 1.034 ± 0.38 *** | 1.209 ± 0.48 *** | 0.899 ± 0.65 ** |
| | LD | 0.234 ± 0.54 * | 0.824 ± 0.44 ** | 0.945 ± 0.48 ** | 0.545 ± 0.799 ** |

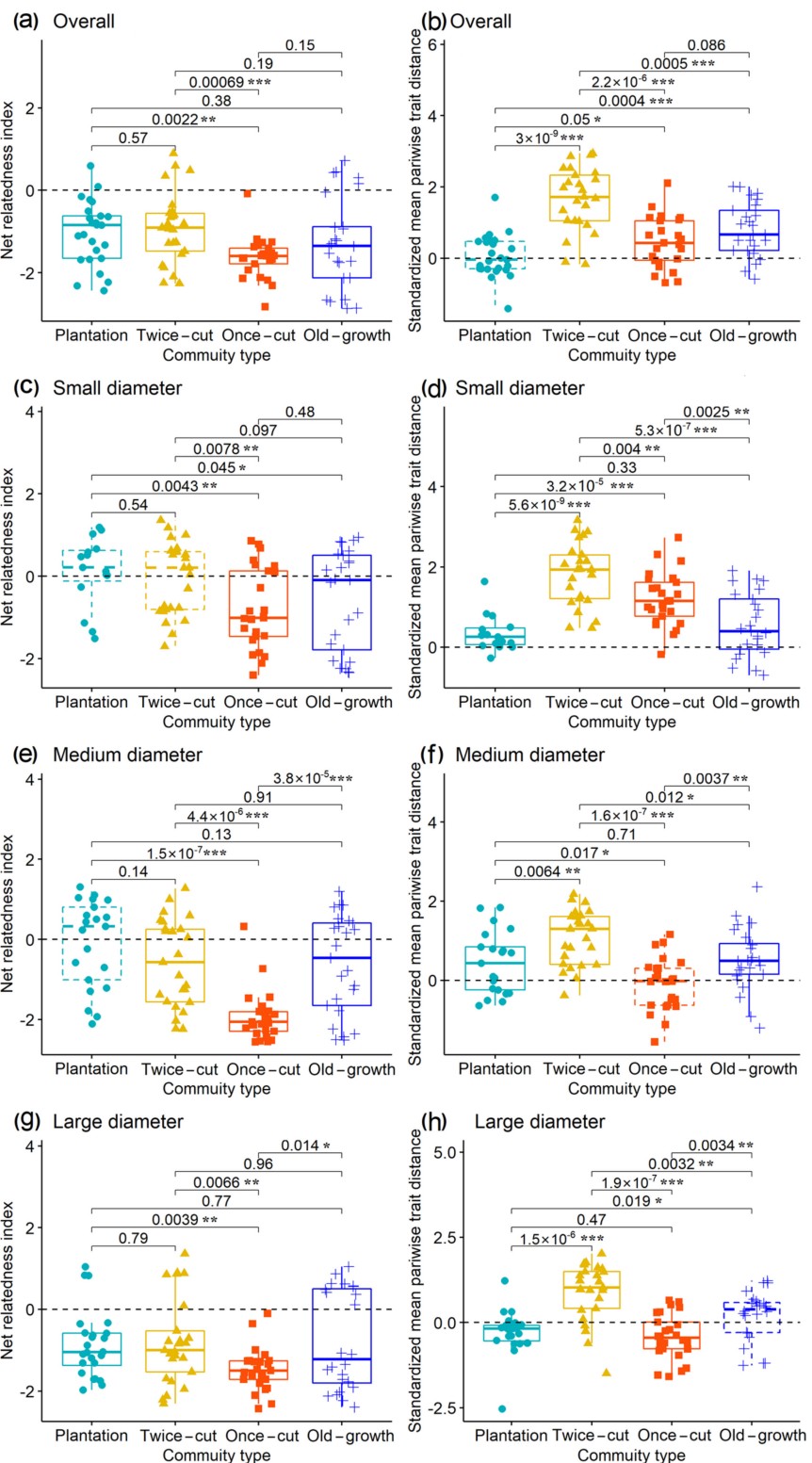

**Figure 4.** Phylogenetic and functional structure of woody plants in the four disturbance communities. Left, net relatedness index (*NRI*) of overall (**a**), small (**c**), medium (**e**), and large (**g**) diameter classes. Right, standardization mean pairwise distance (*S.E.S PW*) of overall (**b**), small (**d**), medium (**f**), and large (**h**) diameter classes. The black dashed lines at 0 indicate no turnover. Bold box lines represent means that are significantly different from 0, while dashed box lines represent non-significance, using *t*−tests. Lines joining boxes show the results of Wilcoxon tests between disturbance regimes ($p \leq 0.05$ level of significance). "*", "**", and "***" represent $p < 0.05$, $p < 0.01$, and $p < 0.001$, respectively.

**Table 5.** Results of *t*-tests of the hypothesis that the mean value of *NRI* or *S.E.S PW* is zero in pairs of different disturbance communities. D1, D2, D3, and D4 represent the plantation, twice-cut, once-cut, and old-growth forests, respectively. "*" and "***" represent $p < 0.05$ and $p < 0.001$, respectively.

| | D1–D2 | | D1–D3 | | D1–D4 | | D2–D3 | | D2–D4 | | D3–D4 | |
|---|---|---|---|---|---|---|---|---|---|---|---|---|
| | Mean | *t* | Mean | *t* | Mean | *t* | Mean | *t* | Mean | *t* | Mean | *t* |
| NRI | | | | | | | | | | | | |
| Overall | 1.96 | 47.15 *** | 2.4 | 63.02 *** | 2.11 | 43.08 *** | 1.75 | 48.82 *** | 1.48 | 27 *** | 2.01 | 40.76 *** |
| SD | 0.34 | 5.25 *** | 0.69 | 11.22 *** | 0.68 | 9.74 *** | 1.03 | 19.59 *** | 1.03 | 15.82 *** | 1.47 | 23.27 *** |
| MD | 0.72 | 9.71 *** | 1.58 | 28.28 *** | 0.83 | 10.52 *** | 1.88 | 47.92 *** | 1.05 | 15.62 *** | 1.86 | 41.18 *** |
| LD | 1.6 | 37.38 *** | 2.09 | 54.57 *** | 1.46 | 23.58 *** | 1.92 | 50.09 *** | 1.21 | 19.99 *** | 1.73 | 30.82 *** |
| S.E.S PW | | | | | | | | | | | | |
| Overall | 1.37 | 41.19 *** | 1.38 | 44.83 *** | 1.52 | 52.26 *** | 0.56 | 15.02 *** | 0.79 | 21.41 *** | 0.91 | 25.92 *** |
| SD | 0.27 | 3.87 *** | 0.3 | 3.86 *** | 0.46 | 7.24 *** | −0.26 | −5.56 | 0.16 | 4.48 *** | 0.01 | 2.10 * |
| MD | 0.9 | 21.40 *** | 1.02 | 26.58 *** | 0.89 | 18.78 *** | 1.29 | 48.11 *** | 1.07 | 33.0 *** | 1.22 | 38.79 *** |
| LD | 0.91 | 31.61 *** | 0.76 | 25.13 *** | 0.81 | 20.61 *** | 0.95 | 32.61 *** | 0.75 | 20 *** | 0.9 | 22.58 *** |

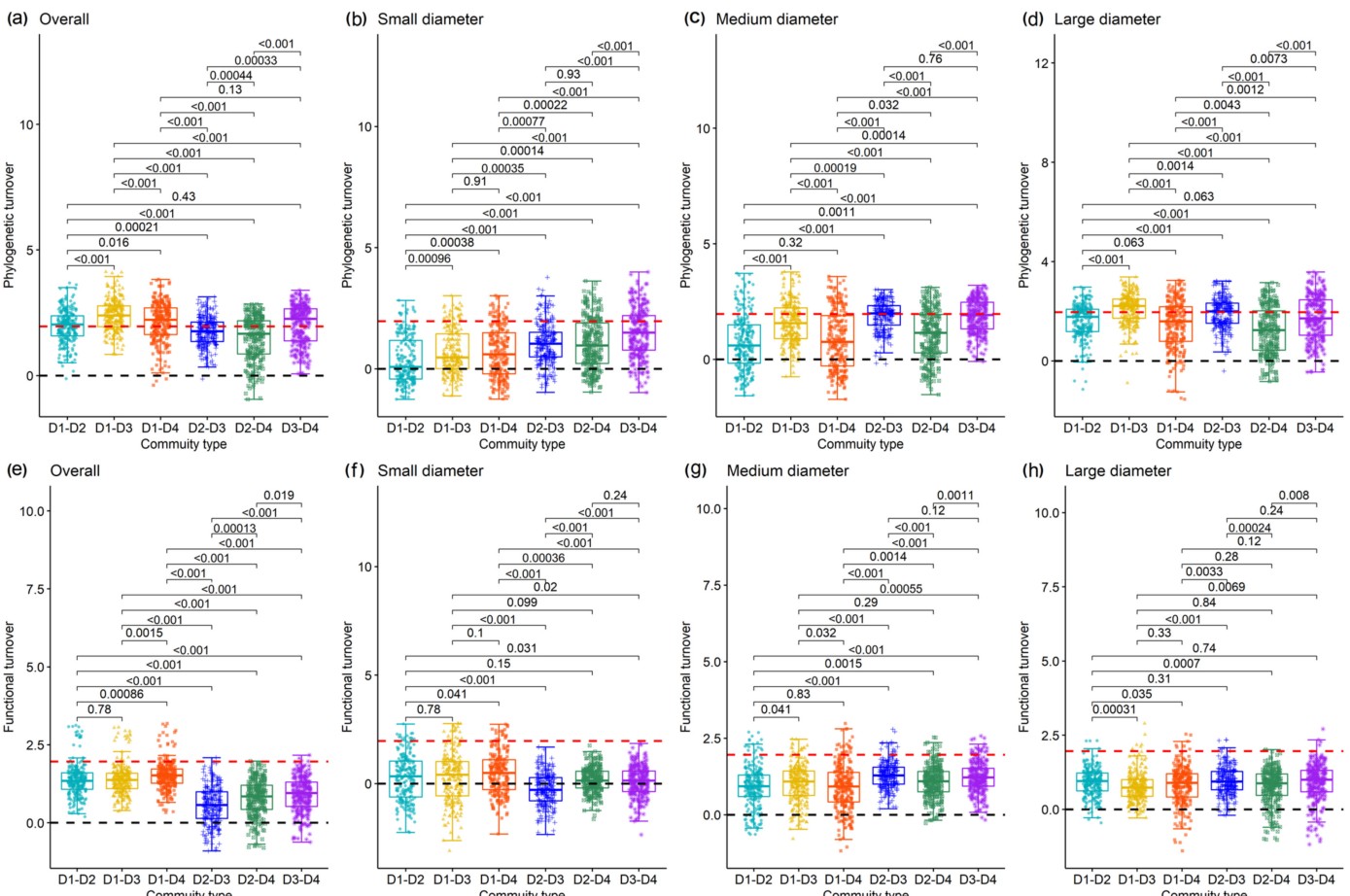

**Figure 5.** Phylogenetic (**a–d**) and functional (**e–h**) turnover between the four disturbance regimes; (**a**,**e**) are the overall diameter species, (**b**,**f**) are the small diameter species, (**c**,**g**) are the medium diameter species, (**d**,**h**) are the large diameter species. Black dashed lines indicate turnover = 0. Red dashed lines indicate turnover = 1.96. D1, D2, D3, and D4 represent the plantation, twice-cut, once-cut, and old-growth forests, respectively. Lines joining boxes show the results of Wilcoxon tests between disturbance regimes ($p \leq 0.05$ level of significance).

### 3.5. Phylogenetic and Functional−Environment Relations among Different Disturbance Plots

The Mantel tests showed that the phylogenetic and functional structures were not generally influenced by many spatial and environmental factors. However, all factors except convexity were significantly related to structure in some disturbance regimes at particular DBH scales (Table 6). Phylogenetic and functional structure were correlated

with spatial distance and slope in the twice-cut and once-cut forests ($p < 0.05$, Table 6). Phylogenetic structure was also correlated with elevation in the once-cut and old-growth forests ($p < 0.05$), but functional structure was only correlated with elevation in the twice-cut forest ($p < 0.05$, Table 6). For small diameter species, only functional structure was correlated with spatial distance in the plantation forest, and only elevation in the twice-cut forest ($p < 0.05$, Table 6). For medium diameter species, phylogenetic and functional structure were correlated with slope in the once-cut forest ($p < 0.05$, Table 6). Finally, for large diameter species, phylogenetic and functional structure were correlated with slope in the twice-cut forests ($p < 0.05$, Table 6), and phylogenetic structure was correlated with spatial distance in the twice-cut forest and correlated with elevation in the once-cut and old-growth forests ($p < 0.05$, Table 6).

**Table 6.** Results of Mantel tests of the relationships between phylogenetic and functional structure with spatial and environment variables. D1, D2, D3, and D4 represent the plantation, twice-cut, once-cut, and old-growth forests, respectively. SD, MD, and LD represent small, medium, and large diameter classes, respectively. "*" and "**" represent $p < 0.05$ and $p < 0.01$.

| Distance Matrix | | Phylogenetic Index | | | | Functional Index | | | |
|---|---|---|---|---|---|---|---|---|---|
| | | D1 | D2 | D3 | D4 | D1 | D2 | D3 | D4 |
| | | R | R | R | R | R | R | R | R |
| Overall | Spatial distance (m) | 0.026 | 0.256 ** | 0.2 * | 0.196 ** | −0.018 | 0.13 * | 0.157 * | −0.017 |
| | Aspect | −0.093 | −0.1 | 0.04 | 0.058 | −0.01 | 0.003 | 0.015 | 0.13 |
| | Slope (°) | 0.024 | 0.22 ** | 0.217 ** | 0.069 | −0.012 | 0.135 * | 0.191* | −0.056 |
| | Elevation (m) | 0.044 | 0.078 | 0.276 * | 0.254 ** | 0.05 | 0.194 * | −0.016 | 0.03 |
| | Convexity (°) | 0.101 | −0.03 | 0.277 | −0.058 | 0.019 | −0.018 | 0.087 | −0.032 |
| SD | Spatial distance (m) | −0.088 | −0.058 | −0.086 | 0.012 | 0.202 * | −0.018 | −0.027 | 0.046 |
| | Aspect | 0.073 | −0.031 | 0.005 | 0.051 | 0.018 | 0.038 | 0.105 | 0.073 |
| | Slope (°) | 0.031 | −0.052 | −0.06 | 0.092 | 0.146 | −0.017 | 0.007 | 0.063 |
| | Elevation (m) | −0.076 | −0.071 | −0.122 | 0.067 | −0.098 | 0.219 * | −0.068 | 0.049 |
| | Convexity (°) | 0.024 | 0.0367 | −0.104 | −0.035 | −0.036 | −0.034 | −0.102 | −0.052 |
| MD | Spatial distance (m) | 0.065 | 0.094 | 0.117 | −0.007 | −0.115 | −0.007 | −0.013 | −0.02 |
| | Aspect | 0.015 | −0.054 | 0.139 | 0.099 | −0.041 | 0.067 | 0.014 | 0.052 |
| | Slope (°) | 0.091 | 0.041 | 0.226 ** | 0.083 | −0.043 | −0.056 | 0.124 * | 0.054 |
| | Elevation (m) | 0.102 | 0.003 | 0.07 | 0.086 | −0.163 | 0.011 | −0.023 | −0.053 |
| | Convexity (°) | −0.038 | −0.035 | 0.06 | −0.028 | −0.129 | −0.021 | −0.068 | −0.088 |
| LD | Spatial distance (m) | 0.053 | 0.211 ** | 0.08 | 0.064 | −0.073 | 0.121 | 0.108 | 0.116 |
| | Aspect | −0.031 | −0.082 | 0.013 | 0.061 | −0.041 | 0.047 | −0.063 | 0.178 |
| | Slope (°) | 0.052 | 0.308 ** | 0.039 | −0.028 | −0.132 | 0.185 * | 0.115 | 0.013 |
| | Elevation (m) | −0.047 | 0.039 | 0.198 * | 0.23 * | −0.118 | −0.021 | 0.095 | −0.05 |
| | Convexity (°) | 0.198 | 0.029 | 0.1143 | 0.033 | −0.105 | 0.04 | −0.125 | −0.059 |

The final MRM models showed that different combinations of spatial and environmental variables were correlated with the phylogenetic and functional structure of various diameter classes in different disturbance regimes (Table 7). At the overall DBH level, spatial distance better explained the phylogenetic β-diversity than environmental distance, but environmental distance better explained the functional trait β-diversity in the four disturbance regimes. Conversely, environmental distance explained more variation in phylogenetic β-diversity in small DBH species of the once- and twice-cut forests, as well as large DBH species in the plantation forest. Meanwhile, spatial distance better explained functional β-diversity than environmental distance for small DBH species in the plantation forest and large DBH species of the once- and twice-cut forests (Table 6).

**Table 7.** Results of multiple regression on distance matrices (MRM) of phylogenetic β-diversity, as predicted by environmental and spatial distance variables in different disturbance communities. D1, D2, D3, and D4 represent plantation, twice-cut, once-cut, and old-growth forests, respectively. M.E.S., multiple regression on distance matrices of environment and space; M.S., multiple regression on distance matrices of space; M.E., multiple regression on distance matrices of environment; M.P.S., multiple regression on distance matrices of pure space; M.P.E., multiple regression on distance matrices of pure environment.

| Explanatory Variable | Phylogenetic Beta Diversity | | | | Functional Beta Diversity | | | |
|---|---|---|---|---|---|---|---|---|
| | D1 | D2 | D3 | D4 | D1 | D2 | D3 | D4 |
| Overall species | | | | | | | | |
| M.S.E | 0.0283 | 0.0680 | 0.0336 | 0.0300 | 0.0289 | 0.0449 | 0.0260 | 0.0294 |
| M.S | 0.0164 | 0.0659 | 0.0191 | 0.0238 | 0.0095 | 0.0100 | 0.0164 | 0.0003 |
| M.E | 0.0003 | 0.0000 | 0.0000 | 0.0126 | 0.0289 | 0.0265 | 0.0001 | 0.0288 |
| M.P.S | 0.0279 | 0.0680 | 0.0336 | 0.0126 | 0.0296 | 0.0353 | 0.0259 | 0.0006 |
| M.P.E | 0.0121 | 0.0023 | 0.0147 | 0.0163 | 0.0000 | 0.0190 | 0.0098 | 0.0291 |
| Small diameter | | | | | | | | |
| M.S.E | 0.0292 | 0.0093 | 0.0139 | 0.0304 | 0.0781 | 0.0884 | 0.0605 | 0.0566 |
| M.S | 0.0150 | 0.0034 | 0.0068 | 0.0267 | 0.0410 | 0.0243 | 0.0003 | 0.0125 |
| M.E | 0.0010 | 0.0040 | 0.0139 | 0.0005 | 0.0022 | 0.0466 | 0.0377 | 0.0318 |
| M.P.S | 0.0282 | 0.0054 | 0.0001 | 0.0300 | 0.0760 | 0.0438 | 0.0237 | 0.0256 |
| M.P.E | 0.0144 | 0.0059 | 0.0072 | 0.0038 | 0.0386 | 0.0657 | 0.0602 | 0.0446 |
| Medium diameter | | | | | | | | |
| M.S.E | 0.0418 | 0.0096 | 0.0131 | 0.0067 | 0.0251 | 0.0062 | 0.0626 | 0.0340 |
| M.S | 0.0047 | 0.0090 | 0.0129 | 0.0064 | 0.0133 | 0.0013 | 0.0425 | 0.0143 |
| M.E | 0.0155 | 0.0000 | 0.0070 | 0.0000 | 0.0240 | 0.0038 | 0.0541 | 0.0270 |
| M.P.S | 0.0267 | 0.0096 | 0.0061 | 0.0067 | 0.0011 | 0.0025 | 0.0090 | 0.0072 |
| M.P.E | 0.0373 | 0.0006 | 0.0001 | 0.0003 | 0.0120 | 0.0049 | 0.0210 | 0.0200 |
| Large diameter | | | | | | | | |
| M.S.E | 0.0146 | 0.0594 | 0.0022 | 0.0447 | 0.0496 | 0.0341 | 0.0430 | 0.0147 |
| M.S | 0.0045 | 0.0449 | 0.0022 | 0.0361 | 0.0076 | 0.0302 | 0.0426 | 0.0023 |
| M.E | 0.0021 | 0.0055 | 0.0010 | 0.0130 | 0.0478 | 0.0007 | 0.0171 | 0.0143 |
| M.P.S | 0.0126 | 0.0542 | 0.0013 | 0.0089 | 0.0019 | 0.0335 | 0.0264 | 0.0005 |
| M.P.E | 0.0101 | 0.0152 | 0.0000 | 0.0321 | 0.0423 | 0.0041 | 0.0004 | 0.0124 |

Generally, as the disturbance intensity decreased (and as forest age increased), the explanatory power of spatial distance for phylogenetic β-diversity structure decreased and that of environmental distance on phylogenetic structure increased. Furthermore, the spatial and environmental distances had the largest explanatory power for phylogenetic and functional β-diversity, respectively, in the moderate disturbance communities, and the smallest in the undisturbed communities.

## 4. Discussion

### 4.1. Phylogenetic Signals of Functional Traits

Determining the degree to which functional traits are evolutionarily conserved is a necessary step in the inference of species coexistence mechanisms [66]. Here, we measured the phylogenetic signals, as measured by Blomberg's *K*, in leaf, stem, and physiological traits of tree species, across disturbance regimes on Baiyun Mountain. Maximum tree height (MTH) and wood density (WD) had relatively weak phylogenetic signals, which may stem from the ubiquitous need of forest trees to grow taller and access higher light environments and as species with higher woody density can support a greater plant height [67]. Whereas non-photochemical quenching (NPQ) is a physiological trait related to chlorophyll fluorescence, which may be less affected by environmental differences and, thus, has a relatively high phylogenetic signal. All functional traits except transpiration rate (TR) in twice-cut forests showed a phylogenetic signal ($p < 0.05$, Table 3). Thus, the functional traits in the Baiyun Mountain forests tended to be evolutionarily conserved [59]; that is, species with similar

genetic relationships had similar functional traits in Baiyun Mountain [68]. Our results are consistent with studies of the Changbai Mountains [69], Gutian Mountains [70], and many other forests around the world [71]. A strong phylogenetic signal may suggest environmental filtering [16,20], while over-dispersion can indicate competitive exclusion during community construction [72]. By combining patterns of community functional traits and phylogenetic structure it is possible to assess the causes of community construction [68].

### 4.2. Community Phylogenetic and Functional Structure

The phylogenetic and functional traits of the overall diameter class showed a non-random structure at different spatial scales, with significant β-diversity in community phylogenetic and functional traits in all disturbance regimes of the Baiyun Mountain deciduous broad-leaved forest (Table 4, Figure S1). This was not consistent with the predictions of neutral theory [27], and rather supports the notion that niche processes play an important role in community construction in this deciduous broad-leaved forest, regardless of the disturbance regime.

We found that small diameter species showed a random or slightly over-dispersed phylogenetic structure, and that over-dispersion increased significantly with diameter class. The diameter class of plants can be taken as a proxy for forest age [73]. This suggests that the growth of young trees is relatively phylogenetically clustered, perhaps due to dispersal limitations, but as individuals grow and compete, only a small number of large trees persist within communities at greater mean geographical distances [74]. This is consistent with previous findings that the phylogenetic structure of small diameter trees tends to be clustered or random, while that of large diameter trees tends to be over-dispersed [20]. We observed similar trends in functional structure, which suggests that competitive exclusion plays a major role in community construction at the large diameter size scale in the Baiyun Mountain deciduous broad-leaved forest, regardless of the disturbance regime.

### 4.3. Ecological Processes of Community Construction in Different Disturbance Regimes

We found significant differences in community phylogeny and functional trait structure among the different disturbance regimes in Baiyun Mountain deciduous broad-leaved forest, which indicates that the ecological processes of community construction are likely also different. Most human-disturbed forests are in the early or middle stages of succession [35], when pioneer trees play a dominant role, due to having small seeds, wide propagation, fast growth, and strong plasticity [29]. Early succession communities are often composed of closely related species, and thus moderate to highly disturbed communities tend to exhibit phylogenetic clustering [69]. However, the short life span of pioneer tree species in early succession results in their decline and replacement during forest regeneration [44]. Disturbance theory suggests that moderate disturbances increase resource availability and species richness [32]. During the later stages of succession, as dispersal is restricted and light becomes less available, competition among species for environmental resources increases, and competitive exclusion becomes a dominant process. Competitive exclusion reduces the immigration of closely related species with similar ecological niches and therefore leads to community over-dispersion [18]. Our results are consistent these aspects of disturbance theory: over-dispersion generally increased in the less disturbed plots [75,76].

The results of variance decomposition by MRM further showed that as the disturbance intensity decreased, spatial distance better explained phylogenetic and functional turnover, while the explanatory power of environmental distance decreased. That environmental distance better explained the phylogenetic and functional trait turnover in moderate to high disturbance communities suggests the importance of habitat filtering in community construction. Moreover, although competitive exclusion is often dominant in less disturbed communities [8], we found that spatial distance had a higher explanatory power of turnover in old-growth forests, consistent with diffusion limitation [27]. In conclusion, as observed in the Changbai Mountain coniferous mixed forest [69], environmental filtering plays a

dominant role in community construction in the early stages of succession in high and moderate disturbance regimes, while competitive exclusion and diffusion limitation become more important in the later stages of succession [75].

Past studies of phylogenetic and functional structure across tree sizes, spatial scales, and disturbance regimes have not been entirely consistent. For example, Mo et al. found phylogenetic clustering in a young, early succession secondary forest, over-dispersion in an old secondary forest, and finally random structure in an old, late succession forest; presumably, the result of habitat filtration and competitive exclusion [77]. Whereas, Yang et al. found that medium diameter tree species showed no phylogenetic or functional structure at a small scale (5 m × 5 m), suggesting that neutral processes may play a role at small scales [68]. However, we found that community phylogenetic and functional trait structures were generally non-random, regardless of the disturbance regime or spatial scale, which is not consistent with neutral theory [4].

Finally, we observed differences in the phylogenetic and functional trait $\alpha$- and $\beta$-diversity of tree species at different spatial and tree diameter scales in our Baiyun Mountain plots. The weak phylogenetic signals in functional traits ($K < 1$) may explain the inconsistent patterns in phylogenetic and functional traits. Some studies have suggested that phylogenetic distance may not be a good representation of ecological differences between species if the traits are highly differentiated [78], and studies of community assembly and species coexistence based solely on phylogenetic information may be misleading [27]. Therefore, it is necessary to combine phylogenetic and functional trait information, as we have done here, to accurately infer community assembly mechanisms [68]. It must also be said that inconsistent patterns of phylogenetic and functional traits may stem from incomplete sampling of taxa and functional traits, such that the observed data do not fully represent the actual ecological niches of species [27,79].

## 5. Conclusions

We examined the phylogenetic signals in leaf, stem, and physiological functional traits of tree species from different disturbance plots in Baiyun Mountain, to assess the mechanisms underlying community construction. We generally found phylogenetic signals—and thus evolutionary conservation—in functional traits, regardless of disturbance regime, diameter class, or spatial scale. Our findings suggest that niche, rather than neutral, processes played a major role in community construction in this deciduous broad-leaved forest. Furthermore, environmental filtering tended to be more important following high and moderate disturbance, and competitive exclusion was more important following slight disturbance and in undisturbed communities.

**Supplementary Materials:** The following supporting information can be downloaded at: https://www.mdpi.com/article/10.3390/f13060896/s1, Figure S1: phylogenetic (a,b) and functional trait (c,d) structure (mean ± SE) at different spatial scales in four disturbance regimes across tree diameter classes scales. Solid markers represent means of NRI or S. E. S. PW that are significantly different from 0 and open markers indicate that the difference was not significantly different from the 0 base in the Wilcoxon test.

**Author Contributions:** Conceptualization, P.L., S.D. and Q.F.; Data curation, J.Z.; Formal analysis, P.Z. and J.Z.; Funding acquisition, S.D.; Investigation, P.L. and X.W.; Methodology, P.L. and X.W.; Project administration, P.L.; Resources, S.D.; Software, P.L.; Supervision, S.D.; Validation, Z.G., P.Z. and S.D.; Visualization, P.L. and J.Z.; Writing—original draft, P.L.; Writing—review and editing, P.L., P.Z. and J.Z. All authors have read and agreed to the published version of the manuscript.

**Funding:** This research was funded by the National Nature Science Foundation of China (#42171091).

**Acknowledgments:** We would like to thank Zhendong Hong, and Pengwei Qiu for help with the research idea. We also thank Ruofan Cao, Lei Guo, and Zhiliang Yuan for help with data processing.

**Conflicts of Interest:** The authors declare no conflict of interest.

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
