# Peer review of "Phylogenetic and Functional Structure of Wood Communities among Different Disturbance Regimes in a Temperate Mountain Forest"

_forests, doi:10.3390/f13060896_

Round 1
Reviewer 1 Report
This paper is very interesting and written very well. Each table and graph explain very well. The motivation and novelty of the work is very strong. I strongly recommend it for publication without any changing.
Author Response
We are very grateful for your review of our manuscript in your busy schedule, and we also appreciate your recognition and affirmation of our research. The whole manuscript also has been carefully examined once again. A major professional scientific editing company has been employed to check the English language and spell check of the manuscript again.
Reviewer 2 Report
Manuscript “Phylogenetic and functional structure of wood communities among different disturbance regimes in a temperate mountain forest” by Li et al, assessed the mechanisms underlying community construction using phylogenetic signals in leaf, stem, and physiological functional traits of tree species.
It is interesting manuscript for “Forests”, however, there are few comments that should be addressed by authors before possible acceptance of the manuscript.
Comments:
Keywords: The same words have been applied in Title and Keywords.
Introduction:
Line 38: “these processes”. Which processes?
Line 39: “relative slow” or relatively slow?
Lines 41-42: Ambiguous sentence: “Random phylogenetic …and competitive exclusion”.
Line 47: “improvements” or improved?
Lines 50-51: … community construction, based on … β-diversity, …
Line 52: study scales. Which scales? Please specify.
Line 59: Whereas, …
Lines 63-65: Why most studies do not combine phylogenetic and functional trait diversity?
Lines 89-90: This part is related to M&M not Introduction.
Lines 95-98: Please mention the hypothesis of your research.
Materials and Methods:
Lines 105-106: Mean annual temperature was … minimum and maximum temperatures were …
Line 146: Please mention the “eight key functional traits”
Author Response
Thank you for the very valuable comments on our manuscript, which have greatly helped us to further strengthen our manuscript. All suggested corrections and comments have been modified based on your suggestions. The whole manuscript also has been examined and modified carefully. The content modified has been marked in red in the manuscript. A major professional scientific editing company has been employed to improve the sentence of the manuscript again. We are very grateful for your careful modification of our manuscript.
- Thank you for the very valuable comment. According to the suggestion, we have removed “phylogenetic structure”, “community assembly mechanisms”, and “disturbance regimes” from Keywords (Page 1, Lines 26-27).
- Thanks very much for your pertinent suggestion about Line 38. “These processes” refers to habitat filtering and competitive exclusion processes in niche theory, and diffusion and random processes in neutral theory. We have revised this sentence according to your suggestion. Our manuscript has been revised as “Based on the phylogenetic niche conservation theory of Webb, the phylogenetic distance of species within communities can be used to infer the relative strengths of niche and neutral progresses in community assembly.” (Page 1, Lines 37-38)
- We are very grateful for your careful modification of our manuscript. We have removed “relative slow” and replaced them with “relatively slow” (Page 1, Line 38).
- Thanks for your careful modification of our manuscript. Our original writing did cause ambiguity. We have revised this sentence as “Random phylogenetic structures may result from diffusion and habitat filtering or a combination of random effects and competitive exclusion” (Page 1, Lines 40-42).
- We are very grateful for your careful modification of our manuscript. We have removed “improvements” and replaced them with “improved” (Page 2, Line 46).
- We are very grateful for your careful modification of our manuscript. We have revised this sentence as “… community construction, based on … β-diversity, …” according to your suggestion (Page 2, Line 49).
- Thank you for your constructive comment about Line52. The “study scales” refers to the temporal and spatial scales. Our manuscript has been revised as “Studies have shown that α- and β-diversity of community phylogenetic and functional traits are closely related to study scales both in time and space” (Page 2, Lines 52)
- We are very grateful for your careful modification of our manuscript. According to your suggestion, we have revised this sentence as “Whereas, competitive exclusion is more likely…” (Page 2, Line 59).
- Thank you for your constructive comment about Lines 63-65. The study of phylogenetic and functional trait diversity mainly includes two independent work of plot construction and sampling of functional traits. Perhaps due to the different work arrangements or research plans of other related studies, a few studies may only complete one part of the work, or split the two parts of the work for research separately.
- Thanks very much for your pertinent suggestion. This part (Page 2, Lines 89-90) has been modified and moved to the "Materials and Methods " section. (Page 4, Lines 119-121). Meanwhile, No. 49 and 50 references have been revised accordingly (Page 8, Lines 596-600).
- Thank you for your constructive suggestion. According to your suggestions, we have added the hypothesis of this study to the manuscript (Page 2, Lines 92-99). The hypothesizes include (1) species with similar genetic relationships would have similar functional traits result of significant phylogenetic signals, (2) phylogenetic and functional trait structure in small and medium spatial and DBH class scale could have higher clustered due to habitat filtering and competitive exclusion, and (3) environmental filtering could tend to be more important following high and moderate disturbance due to increase resource availability and species richness, and competitive exclusion could tend to be more important following slight and undisturbed communities due to diffusional limitation.
- We are very grateful for your careful modification of our manuscript. According to your available suggestion, we have revised this sentence as “Mean annual temperature was … minimum and maximum temperatures were …” (Lines 107-108).
- According to your available suggestion, we have added the key functional traits in our manuscript (Page 5, Lines 149-151) “We measured seven key functional traits representing leaf (Specific leaf area, Stomatal conductance), stem (Maximum tree height, Wood density), and physiological (Minimal Fluorescence, Non-photochemical quenching, Transpiration rate) traits of tree species in the community…”
Reviewer 3 Report
Dear authors and editors, The work has merit and a lot of quality, being a great contribution to the area of ​​forest ecology and forest resources. The work is of sufficient quality to be published unmodified.Author Response
We are very grateful for your review of our manuscript in your busy schedule, and we also appreciate your recognition and affirmation of our research. The whole manuscript also has been carefully examined once again. A major professional scientific editing company has been employed to check the English language and spell check of the manuscript again.